# Electrical Field Interactions during Adjacent Electrode Stimulations: eABR Evaluation in Cochlear Implant Users

**DOI:** 10.3390/jcm12020605

**Published:** 2023-01-11

**Authors:** Nicolas Guevara, Eric Truy, Michel Hoen, Ruben Hermann, Clair Vandersteen, Stéphane Gallego

**Affiliations:** 1Institut Universitaire de la Face et du Cou, Centre Hospitalier Universitaire de Nice, Université Côte d’Azur, 06100 Nice, France; 2Department of Audiology and Otorhinolaryngology, Edouard Herriot Hospital, Lyon 1 University, 69437 Lyon, France; 3Clinical Evidence Department, Oticon Medical, 06220 Vallauris, France; 4Institute for Readaptation Sciences and Techniques, Lyon 1 University, 69373 Lyon, France

**Keywords:** cochlear implant, auditory brainstem responses, electrode

## Abstract

The present study investigates how electrically evoked Auditory Brainstem Responses (eABRs) can be used to measure local channel interactions along cochlear implant (CI) electrode arrays. eABRs were recorded from 16 experienced CI patients in response to electrical pulse trains delivered using three stimulation configurations: (1) single electrode stimulations (E11 or E13); (2) simultaneous stimulation from two electrodes separated by one (E_n_ and E_n+2_, E11 and E13); and (3) stimulations from three consecutive electrodes (E11, E12, and E13). Stimulation level was kept constant at 70% electrical dynamic range (EDR) on the two flanking electrodes (E11 and E13) and was varied from 0 to 100% EDR on the middle electrode (E12). We hypothesized that increasing the middle electrode stimulation level would cause increasing local electrical interactions, reflected in characteristics of the evoked compound eABR. Results show that group averaged eABR wave III and V latency and amplitude were reduced when stimulation level at the middle electrode was increased, in particular when stimulation level on E12 reached 40, 70, and 100% EDR. Compound eABRs can provide a detailed individual quantification of electrical interactions occurring at specific electrodes along the CI electrode array. This approach allows a fine determination of interactions at the single electrode level potentially informing audiological decisions regarding mapping of CI systems.

## 1. Introduction

Cochlear implants (CI) are neural prostheses restoring afferent auditory signals in patients with severe-to-profound hearing loss through direct electrical stimulation of the auditory nerve. Despite significant clinical benefits in most users, a large variability of outcomes is observed among recipients. Some CI users may attain near-perfect speech understanding in quiet and high-level performance in noisy environments, while others experience difficulties recognizing simple utterances [1,2,3].

Age at implantation and duration of severe-to-profound loss are regularly identified predictors of clinical performance [2,4,5]. The degree of spiral ganglion neuron (SGN) survival, decreasing with chronological age or deprivation duration, could partly explain interindividual performance differences as it can directly impact the spectral and temporal representation of sound delivered [6,7,8]. Patient performance can also be affected by neural channel interaction, that may originate from intracochlear current diffusion [9,10] and temporal masking or facilitation effects caused by consecutive electrical stimulation of the same neural population [11,12,13,14]. Neural channel interaction limits the number of functionally independent spectral channels available to patients, negatively impacting speech perception [15,16,17]. Developing a clinically relevant method to assess the electrode-neuron interface in CI users is critical to optimize CI fitting methods and improve patient performance.

The quality of CI users’ electrode-neuron interface, in terms of neural channel interaction, can be estimated by different psychophysical and electrophysiological methods such as the psychophysical forward masking [18,19,20,21,22] and the electrically evoked compound action potential (eCAP) forward masking paradigms [23,24,25,26]. Brochier and colleagues (2020) compared different behavioral and electrophysiological measures of neural health in CI users and concluded that different measures likely reflect different properties of the electrode-neuron interface, highlighting the interest of understanding the specific advantages and limitations of the different approaches [27].

We developed an alternative approach to eCAP forward masking, based on the recording of electrically evoked Auditory Brainstem Responses (eABR). Although eCAP recordings have the advantage that they can be acquired directly by the CI system through reversed telemetry, and do not require additional equipment, eABRs can still provide an interesting alternative measurement method by allowing more flexible stimulation and recording paradigms. These different methods therefore appear complementary as they hold the potential for addressing different issues related to neural stimulation by CIs and, together with other behavioral and electrophysiological methods, can enrich the toolset of clinicians [28]. Additionally, the choice of using eABRs was also motivated by the aim of observing ‘local’ interactions occurring in small groups of three neighboring electrodes.

Guevara et al. (2016) developed an eABR-based method to evaluate possible masking effects that are produced by the stimulation of four electrodes distributed along the CI electrode array [17]. They quantified channel interactions as the monoaural interaction component (MIC). MIC was defined and calculated as the ratio of the eABR wave V amplitude obtained in response to a multi-electrode stimulation of the four electrodes and from the sum of eABR traces that were recorded in response to the independent stimulation of each of the four electrodes.

The present work aims to complement the original study by Guevara et al. (2016) [16] and extends the application of MIC to the study of more ‘local’ channel interactions, occurring between three adjacent electrodes.

Using three neighboring electrodes, we recorded eABR traces obtained (1) when stimulating independently from the three individual electrodes, or (2) when stimulating from the two flanking electrodes (E11 and E13), or (3) when stimulating from all three electrodes but varying the stimulation amplitude at E12. Our strategy was to compare the sum of eABR traces obtained from individual recordings in (1) to the recording obtained in (3), depending on the stimulation level at E12. The MIC was defined as the ratio between the sum of eABR wave V amplitudes observed with single electrode stimulations (1) and the amplitude of wave V observed in the three-electrode stimulation condition at different E12 stimulation levels (3). We hypothesized that the MIC would constitute a measure of electrode interaction with MIC values close to one reflecting low interactions (the sum of individual recordings and the sequential recording showing similar amplitudes). On the contrary, higher interaction levels would be indicated by larger MIC values, eABR wave V amplitudes being lower in the sequential stimulation condition than the sum of individual recordings from the same three electrodes. Neural channel interaction to stimulation on three adjacent electrodes was assumed to increase with increasing stimulus intensity at the middle electrode.

## 2. Materials and Methods

### 2.1. Participants and Fitting Method

Sixteen (11 females/5 males, aged between 19 and 81 years old) adult Digisonic SP CI users, fitted with EVO straight lateral-wall electrode array (Oticon Medical, Vallauris, France), with 20 contacts (1: apical, 20: basal), took part in this study. All patients underwent cochlear implantation performed either by a standard posterior tympanotomy approach or by an endaural approach [29,30]. Participants were all native French speakers and have used their implant for at least 10 months prior to the experiment. Duration of deafness, etiology, and duration of CI use were variable, and participants showed a wide range of word recognition performance. Demographic data is presented in Table 1.

### 2.2. Stimuli and Procedure for eABR Recordings

Stimuli were generated by a portable stimulator (Neurelec-Digistim^®^, Oticon Medical, Vallauris, France) that was linked to a PC, via a USB port. Stimulation pulses were the same as those used clinically, i.e., charge-balanced, anodic first “pseudo-monophasic” pulses with a capacitive discharge phase [31]. Pulse amplitude was fixed at 0.9 mA and pulse duration, corresponding to the duration of the leading anodic phase, was varied between 20 µs and 250 µs. The stimulation rate was 47 Hz and the pulse onset asynchrony was adjusted depending on pulse duration. eABR averaging involved 1500 sweeps and the recording duration was 10 milliseconds per sweep. Electrodes (E) numbers 11, 12, and 13, were used for stimulation, with the original clinical patient-specific stimulation parameters (T- and C-levels). Due to the large variation of T- and C-levels across participants, and stimulation level was defined as the percentage of the electrical dynamic range (EDR). Stimulation level on E11 and E13 was always fixed at 70% EDR, and was changed on E12 to be 0, 20, 40, 70, or 100% EDR. eABRs were recorded in response to three stimulus conditions: (1) To electrical stimulation on single electrodes, (2) to sequential electrical stimulation on E11 and E13, and (3) to sequential electrical stimulation on E11, E12, and E13. These stimulation conditions were termed as single, dual, and triple, see Figure 1.

### 2.3. eABR Recordings

During eABR recording sessions, to limit artifacts due to muscle activity, participants were encouraged to relax. Recording sessions lasted 3 to 4 h including breaks. The eABR recording electrode montage consisted of surface electrodes placed on the forehead, the lower part of the chin, and the contralateral mastoid (common). The 10 ms long recordings (512 samples) were synchronized with the stimulus onset and were obtained via the Centor USB (Racia-Alvar^®^, Paris, France) ABR recording system at a 100 µV full-scale range to allow for the online/offline visualization of both the eABR signal and the CI stimulation artifact. Responses were filtered using an analog bandpass filter (1.6 Hz–3.6 kHz) and averaged over 1500 sweeps.

### 2.4. Analysis of eABR Traces and the Monaural Interaction Component (MIC)

The eABR wave III and V peaks and valleys were identified visually based on their expected latencies. Wave III and V peak amplitudes were calculated as the difference (in µV) between the trough and its respective following positive peak for both waves. Peak latencies were calculated from the stimulus onset to the peak of the waveform.

eABR recordings were analyzed based on three types of derived eABR traces that were obtained based on eABR traces to stimuli illustrated in Figure 2. “Independent sum” (IS) was obtained as the sum of three eABR traces recorded in the single stimulus condition (Figure 2a). “Odd sequential” (OS) was obtained as the sum of eABR traces obtained in the dual stimulus condition and the single stimulus condition using E12 (Figure 2b and see recordings A + B in Figure 1). “Continuous sequential” (CS) was defined as the eABR trace recorded in the triple condition (Figure 2c and see recordings (C) in Figure 1).

For each participant, the monaural interaction component (MIC) was defined as the ratio between the peak amplitudes of the wave V components obtained from the sum of individual and the multi-electrode stimulations. The amplitude peak of wave V was selected both for a practical and theoretical issue. In practice, wave V is more robustly observed in patients than wave-III, which can sometimes be harder to identify. In theory, wave V is generated higher in the auditory system (inferior colliculus) than wave-III is (superior olivary complex) and may represent a better candidate to observe a correlation with integrated auditory processes such as speech perception. We hypothesized that the MIC would represent an approximation of the interaction between the concerned electrodes.

### 2.5. Statistical Analyses

Average values for eABR wave III and V peak amplitudes and onsets were determined for each of the 16 participants in the three different calculation methods and for the five different levels of E12 EDR. Repeated-measures ANOVAs were run using the average peak and latency values as dependent variables and eABR calculation method (3 levels: Continuous Sequential, Odd Sequential, or Independent Sum) and E12 EDR (5 levels: 0, 20, 40, 70, or 100%) as factors. The existence of specific effects was further investigated by LSD post-hoc analyses (α ≤ 0.05) when significant main effects or interactions were observed on the two factors.

## 3. Results

Figure 3 shows the across-participant average of the derived eABR traces for different stimulation levels on the middle electrode, E12. These results show that, in general, the MIC increases with increasing stimulation level on the middle electrode: 1.25, 1.01, 1.58, 2.06, and 2.86 for 0, 20, 40, 70, and 100% E12 EDR, confirming an increase in electrical channel interaction.

Figure 4 shows the derived eABR wave III and wave V latency and amplitude results that were analyzed on an individual basis for 11 out of the 16 participants that had clearly visible wave III and wave V in all experimental conditions.

To investigate the effect of eABR latencies and amplitudes for waves III and V, two-way repeated measure analysis of variances (ANOVAs) was conducted. This resulted in four ANOVAs having different dependent variables and a repeated factor design. The two repeated factors corresponded to the derived eABR types (CS, OS, and IS) and E12 EDR (0, 20, 40, 70, and 100% EDR). When significant main effects and interactions were observed, the existence of specific effects was further investigated by LSD post-hoc analyses (α < 0.05).

### 3.1. Latencies

Wave III latencies (Figure 4A) in the derived eABR traces increased from 2.18 to 2.29, and to 2.32 ms as the derived eABR type was changed from CS to OS, and to IS, respectively. ANOVA showed that this factor, the derived eABR type, was marginally significant F(2, 14) = 3.62; *p* = 0.05. Post-hoc analyses showed that latencies were significantly longer when comparing CS to OS (*p* = 0.04), and IS (*p* = 0.03), all other comparisons were not non-significant.

A similar trend can be observed in the wave V latencies (Figure 4B) that increased from 3.84 to 3.94, and to 3.96 ms as the derived eABR type was changed from CS to OS and to IS, respectively. ANOVA did not show any significant effect of derived eABR type on wave V latencies F(2, 20) = 2.50 (*p* = 0.11), however, it showed a significant interaction between the derived eABR type and the stimulation level on the middle electrode F(8, 80) = 2.13 (*p* < 0.05). Post-hoc analysis showed that wave V latency was significantly reduced in CS compared to OS and IS at 40, 70, and 100% EDR on E12 (all *p* values < 0.05 for these conditions). 

### 3.2. Amplitudes

Panel C in Figure 4 shows the wave III amplitudes in the different derived eABR traces at various stimulation levels on the middle electrode. The average wave III amplitude values increased gradually from 259.0 to 316.7, and to 442.7 nV as the derived eABR type was changed from CS to OS, and to IS, respectively. The effect of derived eABR type F(2, 12) = 4.48 (*p* < 0.05) and stimulation level on the middle electrode F(4, 24) = 3.24 (*p* < 0.05) were significant, but no significant interaction was observed F(8, 48) = 1.81 (*p* = 0.09). 

Figure 4D shows that the average wave V amplitudes increased gradually from 735.3, 1003.1, and to 1185.3 nV as the derived eABR type was changed from CS to OS and to IS, respectively. The effect of both the derived eABR type F(2, 20) = 9.29 (*p* < 0.05) and the stimulation level on the middle electrode F(4, 40) = 9.36 (*p* < 0.05) were statistically significant, as well as the interaction F(8, 80) = 3.76 (*p* < 0.05). Post-hoc comparison showed a significantly lower value of wave V amplitude in CS compared to OS and to IS when the stimulation level was 40, 70, and 100% EDR on E12 (all *p* values < 0.05). Wave V amplitude was significantly lower in OS than in IS, only when the stimulation level on E12 was 100% EDR (*p* < 0.05), this effect remained non-significant when the stimulation level on E12 was 0 and 20% EDR.

## 4. Discussion

eABR wave III and V showed lower amplitudes and shorter latencies in the derived eABR type CS compared to OS and IS. Further analysis showed that, in the derived eABR type CS, the wave V latencies were shorter, and amplitudes smaller compared to those in type IS and OS, when the stimulation level on the middle electrode E12 was 40, 70, and 100% EDR.

The derived eABR IS was based on recordings from stimulation on single electrodes, therefore neural channel interaction was not present in the corresponding results. In line with earlier studies, e.g., [32,33,34], eABR wave V amplitudes to increasing stimulation intensity on the middle electrode increased, hence the wave V amplitude in IS also increased. Regarding the latency of wave V, previous studies have reported both an increase [35] and decrease [36] with increasing the stimulus pulse duration. Furthermore, a study by Gallego and colleagues [33] have reported a stimulus duration independent eABR wave V latency. As shown in Figure 4B, wave V latency in IS, obtained from the sum of three eABR traces to independent stimulation on the three electrodes, remained independent of the stimulus pulse duration on the middle electrode, indicating that the wave V latency in IS was likely dominated by a constant wave V latency obtained in response to the stimulation with fixed duration on the flanking electrodes.

The derived eABR OS was based on the sum of recordings to sequential stimulation on the flanking electrodes and the individual stimulation on the middle electrode. The reduction in wave V amplitudes in OS compared to IS likely reflecting the neural channel interaction during the sequential stimulation on the flanking electrodes. The neural population that falls within the overlapping region of the fields of excitation of E11 and E13 may only fire in response to the stimulation on E11 or E13. If a neuron has fired during the first pulse, i.e., stimulation on E11, it is unlikely to fire again during the second pulse, on E13, because the 20 µs inter-pulse interval is less than the absolute refractory period of the auditory neurons [37,38,39,40]. On the other hand, there may have also been neurons for which independent stimulation from E11 and E13 would have been sub-threshold, but due to facilitation effects [37,39,41], these neurons fired in response to the consecutive stimulation. eABR amplitude was previously shown to depend on the spatial extent of neural recruitment by the electrical stimulation e.g., [42], therefore the result that, in general, wave V amplitudes of OS remained lower than those of IS, may be an indication that neural population affected by refractoriness was larger than the population aided by facilitation. Similar to the results for IS, the growth of the wave V amplitude in OS with increasing stimulation level on the middle electrode is caused by the eABR wave V amplitude growth to the independent stimulation on the middle electrode. The wave V latency in OS was likely dominated by the latency that was obtained from the eABR trace in response to the sequential stimulation on the flanking electrodes because it did not change with increasing stimulus duration on the middle electrode. Although, one may expect a latency increase in OS compared to IS due to a temporally extended neural firing during the consecutive stimulation on the flanking electrodes, this increased latency was not visible in the results.

The derived eABR CS was based on recordings to sequential stimulation on three neighboring electrodes. eABR wave V amplitudes in CS did not increase with increasing stimulation level on the middle electrode and were significantly lower than in OS and IS when stimulation level on the middle electrode was 40, 70, or 100% EDR. The field of excitation of the middle electrode, at any stimulation level, may have been completely overlapping with the field of excitation of the flanking electrodes, therefore it is likely that the stimulation from the middle electrode did not increase the number of recruited fibers compared to what could have also been recruited by the flanking electrodes alone, on which the stimulation level was fixed, hence resulting in a flat growth function for CS. Similar to the derived eABR OS, wave V latencies were expected to increase due to a broadening peak therefore the observed decrease in the latency with increasing stimulation level on the middle electrode is an unexpected result with an unclear origin.

The current results indicate that neural population excited by three neighboring channels, in a consecutive manner, can also be recruited by using only the flanking electrodes if the stimulation level on these electrodes is relatively high, i.e., 70% EDR in the current study. This result is in line with the well-known observations by other studies that the number of independent channels in CI users is limited [20,43,44,45].

### Study Limitations

Some limitations pertaining to the present work should be highlighted. The relatively long duration of the measure protocol used in the present experiment (up to four hours per patient), may appear rather long compared to other available electrophysiological measures such as eCAPs for example [24,25], which are fast and may be recorded directly from the cochlear implant. However, this observation should be relativized by the fact that for the current experiment, we used a wide range of measure-configurations and stimulation intensities in order to obtain a full-picture of the approach, but the MIC could be obtained using much fewer stimulation combinations, thereby effectively reducing the recording time to less than an hour and increasing the clinical relevance of the approach. Moreover, eABRs could be selectively used only on a subset of electrodes which would be ‘suspected’ of creating high interactions, for example, after analysis of post-images identifying ill-placed electrodes within the cochlea [46,47,48]. Further work could aim at directly comparing different electrophysiological approaches for the characterization of channel interactions in a systematic way.

## 5. Conclusions

The present study proposes a new eABR measure-based procedure to assess bioelectrical channel interaction among adjacent electrodes along a CI electrode array. Increasing the delivered charge on the middle one of three electrodes, when electrical stimulation is delivered sequentially within the same pulse-train on these electrodes, results in a decrease of the eABR wave III and V latencies and amplitudes. Future research will directly compare different methods of quantifying electrical interactions in CI users and may also test the effect of deactivating electrodes, at which the strongest neural channel interaction is identified by the current method, on patient performance.

## Figures and Tables

**Figure 1 jcm-12-00605-f001:**
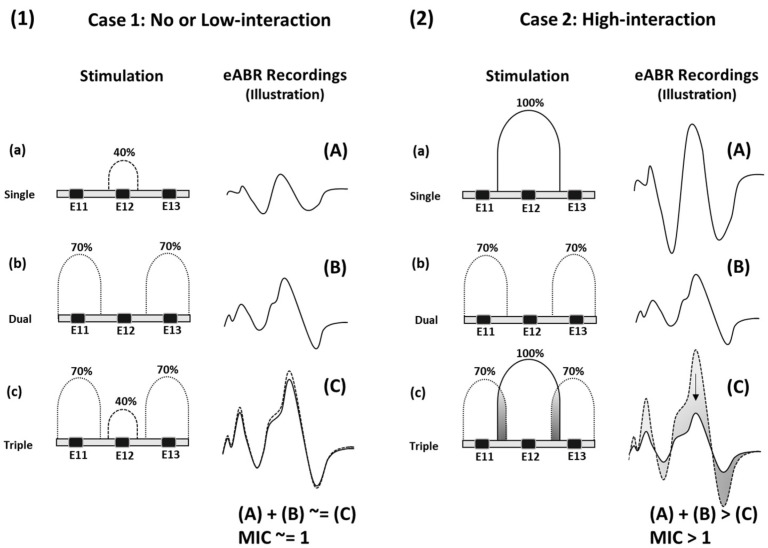
Schematic overview of the stimulation conditions (left side, small letters) and eABR recordings (right side, capital letters) illustrating two hypothetical cases: no or low-interaction—(**1**) left pane, case 1: E12 stimulating at 40% and high-interaction—(**2**) right pane, case 2: E12 stimulating at 100%. In (**1c**): low intensity stimulation at E12 causes no or low interactions, the recording (**1C**) (solid line) is close or equal to the sum of recordings (**1A**,**1B**) (dashed line), obtained when stimulating from the mid-electrode (**1a**) or flanking electrodes (**1b**) only, the MIC value is close to 1. In (**2c**): high intensity stimulation at E12 causes high interactions, the recording (**2C**) (solid line) is lower in amplitude than the sum of recordings (**2A**,**2B**) (dashed line), obtained when stimulating from the mid-electrode (**2a**) or flanking electrodes (**2b**) only, the MIC value is increasing. Intensities are expressed in % of the clinical electric dynamic range (EDR) of the electrode. Grey shaded areas represent electrical interactions in (**2c**) and their effect on the eABR recording in (**2C**).

**Figure 2 jcm-12-00605-f002:**
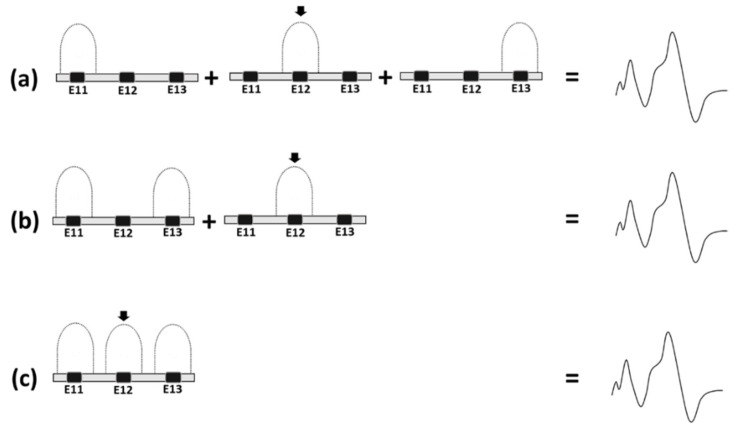
Schematic representation of the eABR manipulations. (**a**) Independent Sum; (**b**) Odd Sequential; and (**c**) Continuous Sequential. Black arrows show the electrode stimulating at varying intensity levels.

**Figure 3 jcm-12-00605-f003:**
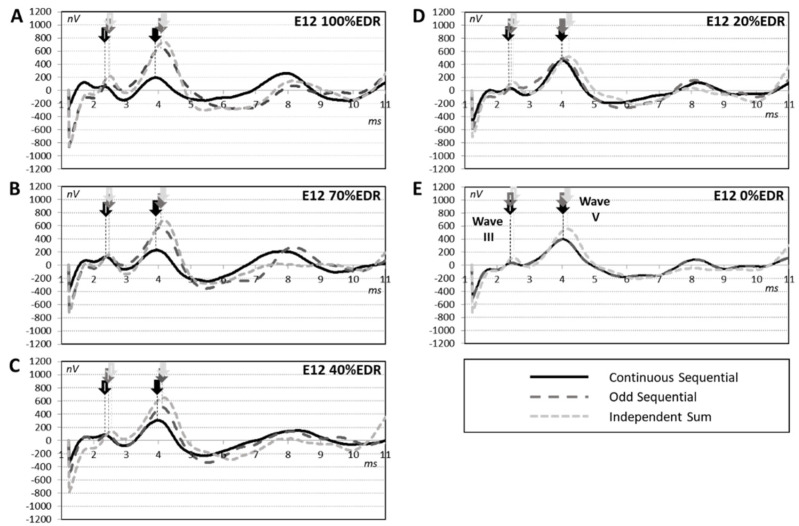
eABR traces combinations averaged across participants. Each trace represents a condition; dark lines represent “Continuous Sequential (CS)”, dark-grey dashed lines represent “Odd Sequential (OS)” and light-grey dashed lines represent “Independent Sum” (IS). The different panels (**A**–**E**) show the results of conditions when varying E12 EDR from 100% (**A**) to 0% (**E**). Unfilled arrows represent wave II while filled arrows represent wave V. Arrows indicate wave III and wave V (**E**), colors follow the same color-codes as eABR traces.

**Figure 4 jcm-12-00605-f004:**
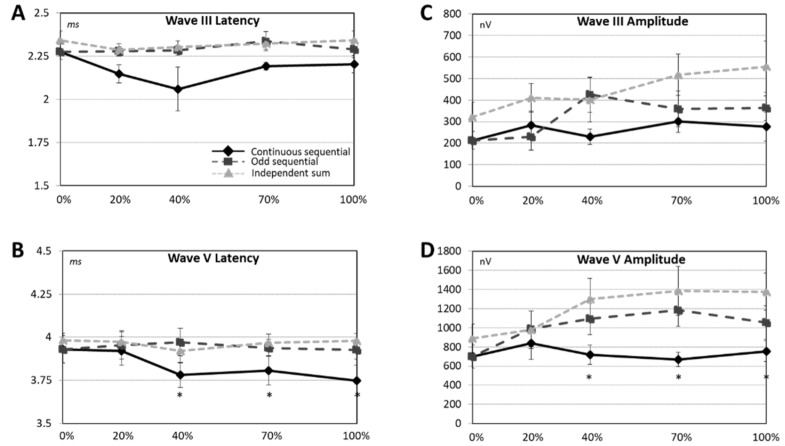
Summary of averaged eABR across subjects and conditions CS, OS, and IS. Wave III and V latencies are shown in panels (**A**,**B**) while their respective amplitudes are shown in panels (**C**,**D**). The color code matches those of Figure 3. The asterisks represent the amplitudes showing significant difference during the post hoc analysis.

**Table 1 jcm-12-00605-t001:** Demographic details of included patients, Gender: f: female, m: male; VCV test in quiet: percent correct VCV logatome identification score in Quiet.

Patient	Gender	Age (Years)	Deafness Duration (Years)	CI Experience (Years)	Etiology	VCV Test Quiet	eABR Waves III and V Identified
1	f	48	8	5	Congenital	12.5%	no
2	m	59	37	11	Otosclerosis	18.7%	no
3	f	81	31	12	Unknown	18.7%	no
4	m	62	16	0.85	Otosclerosis	21.9%	yes
5	f	61	17	3	Unknown	23.4%	yes
6	f	67	24	6	Unknown	26.6%	yes
7	f	55	3	5	Congenital	28.1%	yes
8	m	58	18	1.5	Congenital	37.5%	yes
9	f	19	0	12	Congenital	45.3%	yes
10	m	46	37	1.2	Otosclerosis	53.1%	no
11	f	38	3	4	Congenital	54.7%	yes
12	f	65	39	7	Unknown	56.2%	yes
13	f	48	37	5	Unknown	64.1%	yes
14	f	55	35	4	Otosclerosis	70.3%	yes
15	f	49	29	2.5	Unknown	73.4%	no
16	m	52	44	1.5	Unknown	75.0%	yes
Count or Mean (S.D.)	11 f, 5 m	53.9 (13.7)	23.6 (14.6)	5.1 (3.7)	-	42.5 (21.6)	-

## Data Availability

Not applicable.

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
