# Peer review of "Electrical Field Interactions during Adjacent Electrode Stimulations: eABR Evaluation in Cochlear Implant Users"

_jcm, 2023, doi:10.3390/jcm12020605_

Round 1

Reviewer 1 Report (Previous Reviewer 1)

After the corrections the manuscript is ready for pubblication

Author Response

We would like to thank reviewer 1 for their positive feedback on our manuscript.

Reviewer 2 Report (New Reviewer)

The article is well written, sound scientific and may have an important clinical applications. 

However number of points must be clarified. 

1.       The statement that the whole recording took up to 4 hours  makes it very difficult to clinical application,  giving the short duration using the eCAP.

2.       How the authors  rule out the effect that the effect of high-interaction is not simply effect of high level of stimulation?? Did they have reports from the participant regarding the loudness in each of the paradigm of stimulation??

3.       Is this approach can be applied for all CI brands??

4.        Why the authors selected  the mid electrodes 11,12,and 13??

5.       Latency and amplitude of eABR may be affected by brainstem neuroplasticity. Giving  the very low VCV scores of some of the participants , how this might affect the conclusion and the results?

6.       List of limitation of this study should be listed. 

Author Response

  1. The statement that the whole recording took up to 4 hours  makes it very difficult to clinical application, giving the short duration using the eCAP.

Authors Answer:

We would agree with Reviewer 2, that a 4-hours recording duration would make clinical applicability of our paradigm, especially compared with other faster methods such as eCAPs, very limited. However, the data presented in the current paper constitute a very thorough and global evaluation of the proposed approach, using many different stimulation configurations and intensities. However, all these configurations do not need to be tested at once to obtain a good estimation of the MIC and the recordings using our approach can be easily simplified to be executed in less than one hour. This point is now thoroughly discussed in the newly added 4.1. Study limitations section.

  1. How the authors rule out the effect that the effect of high-interaction is not simply effect of high level of stimulation?? Did they have reports from the participant regarding the loudness in each of the paradigm of stimulation??

Authors Answer:

We would like to thank Reviewer 2 for the good suggestion, we agree it would have been interesting to obtain loudness judgments on the different stimulation configurations obtained, due to an already long experiment, this could not be done. During the study however, the stimulations levels were adjusted depending on the patient specific electrical stimulations dynamics and chosen to be within the clinical electrical dynamic range (EDR) of each electrode. EDR is defined between T-level: the smallest electrical stimulation leading to a percept (threshold) and C-level: the largest stimulation level before discomfort. The flanking electrodes were stimulating at 70% of the EDR, meaning well withing the comfortable stimulation range of patients, adding the middle electrode stimulation between 0% to 100% of its EDR is comparable to what occurs in normal clinical simulations, leading to a loud but still comfortable stimulation. The dependency between perceived loudness or even sound quality and the MIC or other ABR-derived estimations of channel interaction could be the object of future research.    

  1. Is this approach can be applied for all CI brands??

Authors Answer:

Yes, the same approach if perfectly applicable to all CI brands.

  1. Why the authors selected  the mid electrodes 11,12,and 13??

Authors Answer:

A paragraph was added to the Methods section as requested, summarizing statistical analyses performed, we reproduce this paragraph below for convenience, following a comment from Reviewer 2, we also added some details about the determination of the monaural interaction component (MIC) value in Methods, paragraph e. Analysis of eABR traces and the monaural interaction component (MIC)

  1. Latency and amplitude of eABR may be affected by brainstem neuroplasticity. Giving  the very low VCV scores of some of the participants, how this might affect the conclusion and the results?

Authors Answer:

In our approach, patients serve as their own controls, as differential measures in different conditions are performed and compared within subjects, typically other confounding factors such as individual neurophysiological characteristics of the auditory brainstem will not impact those within-subject comparisons.      

  1. List of limitation of this study should be listed. 

Authors Answer:

Following Reviewer 2 suggestion, we have now added a 4.1 Study limitations section to the discussion of the manuscript (Page 9, line 301). We do now review the limitations of the study in this paragraph, including the ones suggested above by Reviewer 2.

Reviewer 3 Report (New Reviewer)

The authors presented a novel eABR procedure to access the interaction between adjacent electrodes of CI. They found that an increase of current being delivered to the middle electrode will lead to a decrease of the eABR wave III and V latencies and amplitudes. The results of this paper are overall solid.

The authors may address the following minor comments:

Perhaps to label the ARB peaks with index so that the authors can follow.

Page 6, line 180. Why the MIC is 1.25 for 0% E12 EDR, which is even higher than 20% EDR?

In real CI application, what are the typical electrical dynamic range? This study designed three stimulation patterns, CS, OS, and IS. How representative these patterns are?

Is there a model derived from the clinical data? For example, how to predict the EDR based on the recorded eARB? And how accurate the prediction is?

Author Response

  1. Perhaps to label the ARB peaks with index so that the authors can follow.

Authors Answer:

Arrows on Figure 3 indicate waves III and V, an explicit label was added to Figure 3-E and a mention was added to the legend of Figure 3.

  1. Page 6, line 180. Why the MIC is 1.25 for 0% E12 EDR, which is even higher than 20% EDR?

Authors Answer:

We would like to thank Reviewer 3 very much for pointing to this important issue. The MIC is an estimation of channel interaction based on several complex measures, so variations can appear in it exact value, and there is not necessarily a strict linear relationship between recording parameters and the specific value of MIC, the difference between 1.25 and 1.01 is probably non-significant in this context.  

  1. In real CI application, what are the typical electrical dynamic range? This study designed three stimulation patterns, CS, OS, and IS. How representative these patterns are?

Authors Answer:

The stimulation patterns used in our study are representative of what happens during daily clinical stimulations with cochlear implants. There are large interindividual and interelectrode variations in EDRs. The electrical dynamic range used in the study were the electrical dynamic ranges used by the patients for their regular cochlear implant use. Clinical EDRs are defined between T- (threshold) and C-(comfort) levels, hence using a proportion of the patients clinical EDRs (0, 20, 40, 70%) is very representative of what could happen during normal cochlear implant stimulations, where electrical stimulation will typically occur between T- and C- levels from 0 to 100% of the electrode EDR.  

A specific mention regarding the use of original clinical stimulation parameters was added page 3, lines 114-115: Electrodes (E) number 11, 12 and 13, were used for stimulation, with the original clinical patient-specific stimulation parameters (T- and C-levels).

  1. Is there a model derived from the clinical data? For example, how to predict the EDR based on the recorded eARB? And how accurate the prediction is?

Authors Answer:

We would like to thank Reviewer 3 for this really interesting suggestion, it would indeed be very interesting to use the data to model some aspects of the individual patient electrical stimulation. However, the interindividual variability and relative scarcity (N=16) of the present data prevents from using the data in robust models, more observations would be needed to lead to robust-enough models.

This manuscript is a resubmission of an earlier submission. The following is a list of the peer review reports and author responses from that submission.

Round 1

Reviewer 1 Report

The authors wrote an article about Electrical Field Interactions During Adjacent Electrode Stimulations using eABR Evaluation in Cochlear Implant users. The article is well written, very interesting and the topic is hot. I give some suggestion to improve the scientific sound of the article and give a better contribution in literature.

1. In the methods, describe the kind of surgery performed (Posterior timpanotomy; Endomeatal approach; soft surgery etc...). You can use this reference: Freni F, Gazia F, Slavutsky V, Scherdel EP, Nicenboim L, Posada R, Portelli D, Galletti B, Galletti F. Cochlear Implant Surgery: Endomeatal Approach versus Posterior Tympanotomy. Int J Environ Res Public Health. 2020 Jun 12;17(12):4187. 

2. Please in the final part of the methods, use a paragraph to summarize the statistical analysis. 

Author Response

Review Round 1

Authors’ Answers to reviewers

Reviewer 1:

The authors wrote an article about Electrical Field Interactions During Adjacent Electrode Stimulations using eABR Evaluation in Cochlear Implant users. The article is well written, very interesting and the topic is hot. I give some suggestion to improve the scientific sound of the article and give a better contribution in literature.

  1. In the methods, describe the kind of surgery performed (Posterior timpanotomy; Endomeatal approach; soft surgery etc...). You can use this reference: Freni F, Gazia F, Slavutsky V, Scherdel EP, Nicenboim L, Posada R, Portelli D, Galletti B, Galletti F. Cochlear Implant Surgery: Endomeatal Approach versus Posterior Tympanotomy. Int J Environ Res Public Health. 2020 Jun 12;17(12):4187. 

Authors Answer:

Following the suggestion of Reviewer 1, we have added details about the surgical technique used for cochlear implantation in the patient pool of our study. The following sentence was added to the Methods section and a reference to the proposed article by Freni et al. included:

All patients underwent cochlear implantation performed either by a standard posterior tympanotomy approach or by an endaural approach (Freni et al., 2020; Vandersteen et al., 2016).

Added References:

Freni, F., Gazia, F., Slavutsky, V., Scherdel, E. P., Nicenboim, L., Posada, R., Portelli, D., Galletti, B., & Galletti, F. (2020). Cochlear Implant Surgery: Endomeatal Approach versus Posterior Tympanotomy. International journal of environmental research and public health, 17(12), 4187.

Vandersteen C, Demarcy T, Roger C, Fontas E, Raffaelli C, Ayache N, Delingette H, Guevara N. Impact of the surgical experience on cochleostomy location: a comparative temporal bone study between endaural and posterior tympanotomy approaches for cochlear implantation. Eur Arch Otorhinolaryngol, 273:2355-61.

  1. Please in the final part of the methods, use a paragraph to summarize the statistical analysis. 

Authors Answer:

A paragraph was added to the Methods section as requested, summarizing statistical analyses performed, we reproduce this paragraph below for convenience, following a comment from Reviewer 2, we also added some details about the determination of the monaural interaction component (MIC) value in Methods, paragraph e. Analysis of eABR traces and the monaural interaction component (MIC)

  1. Statistical analyses

Average values for eABR wave III and V peak amplitudes and onsets were determined for each of the 16 participants in the three different calculation methods. and for the five different levels of E12 EDR. Repeated-measures ANOVAs were run using the average peak and latency values as dependent variables and eABR calculation method (3 levels: Continuous Sequential, Odd Sequential, or Independent Sum) and E12 EDR (5 levels: 0, 20, 40, 70 or 100%) as factors. The existence of specific effects was further investigated by LSD post-hoc analyses (α ≤ 0.05) when significant main effects or interactions were observed on the two factors.  

Reviewer 2 Report

This manuscript by Guevara et al. describes a paradigm and results to explore electrode interactions using eABRs measured from cochlear implant users.  It seems like the results are interesting but the paper is difficult to read and the figures and their legends need improvements.  

Early in the manuscript, the reader would benefit from an overall explanation of the strategy, something like:  We measured 1) the response to x, 2) the response to y, and 3) the response to x+y together .  Our strategy was to subtract the sum of 1+2 from 3.  If there was a difference, we assume that the difference results from electrode interaction.

Then the manuscript should standardize terminology, and a single term used always for each.  Does “sequential stimulation on E11 and E13” have the same meaning as “the dual stimulus condition” (Abstract)? Does “sequential stimulation of E11, E12, and E13”

have the same meaning as “the triple stimulus condition” (Abstract)? The following sentence is especially confusing:  “Odd sequential” (OS) was obtained as the sum of eABR traces obtained in the dual stimulus condition and the singe (sic) stimulus condition using E12”. As written, the paper is hard to understand.

In the Abstract, the “Conclusions” should be more than the development of a paradigm.  What was observed from using that paradigm and how did it advance knowledge of electrode interactions?

The Methods say that eABR waveform peaks III and V were measured.  How were the peaks determined?  Two successive peaks are clear on Figure 3, but then where is peak IV? 

The Methods say “Stimulation rate was 47 Hz, and, in pulse trains, the inter-pulse interval was 20 uS”.  Please make it clear when pulse single pulses vs. pulse trains were used.  

The Methods say, “The electrode insertion angles were estimated to be 171, 191, and 213 degrees – why are there 3 values?  Is this for 3 different patients?

The Results begins (2nd sentence) with “the MIC increases”.  MIC is not defined and I cannot figure out what it is.

The Results are difficult to understand, because the first figure (Fig. 3) shows individual traces from one subject with the first paradigm, and then the second figure (Fig. 4) jumps to overall results of averaged traces from all subjects to a different paradigm. A more logical flow would make the reading less difficult.  

Figure 1: This figure is important but difficult to understand.  Is the x-axis time or distance?  If the stimuli are “sequential” then the stimulus on electrode E11 occurs before the stimulus on electrode E13 and the axis is time.  But the figure seems to be arranged as the x-axis is space and the stimuli occur simultaneously.  In the  legend, the following  sentence is unclear:  “The single and Triple combination shows an example depicting an increasing in dynamic range 137 (for 20 and 70% EDR). “  The figure’s “single” has a only one trace so how can it be an increasing dynamic range?  And the figure’s triple has 20% and 100% below the traces so how can that be 20% and 70%?  What do the “%” signs mean?

Figure 2: The right panel is confusing – do the lower traces result from E11 only and E12 only and E13 only being stimulated?

Figure 3 legend: Should the legend read “visual designation” (instead of “visual assessment”)? And it is mentioned that there is “bump” at around 6 ms, but that is not visible in the traces.

Discussion:  It is not clear that the first sentence is true – small waveform peaks can be greatly affected whereas large waveform peaks might be invariant. That sentence is not important for this paper and should be cut. The Discussion should begin with the “big-picture” items. 

One big-picture item that should be addressed in the Discussion is estimates of electrode interactions previously by others using different techniques (e.g., forward masking).  Do the author’s results indicate bigger or smaller than previous estimates?

Author Response

Review Round 1

Authors’ Answers to reviewers

Reviewer 2 comments:

This manuscript by Guevara et al. describes a paradigm and results to explore electrode interactions using eABRs measured from cochlear implant users.  It seems like the results are interesting, but the paper is difficult to read and the figures and their legends need improvements.  

Early in the manuscript, the reader would benefit from an overall explanation of the strategy, something like:  We measured 1) the response to x, 2) the response to y, and 3) the response to x+y together .  Our strategy was to subtract the sum of 1+2 from 3.  If there was a difference, we assume that the difference results from electrode interaction.

Authors Answer:

We would like to thank Reviewer 2 for his/her thorough review of our manuscript and for the constructive feedbacks provided on the original version. We hope we could answer the issues raised in a satisfying way. We have added / modified the last paragraph of the introduction in order to answer this first concern. Also partially addressing comment N°6 on the MIC, we specified its definition and calculation mode already in this new paragraph. We reproduce the modified section below for convenience.

The present work aims to complement the original study by Guevara et al. (2016) [18] and extends the application of MIC to the study of more ‘local’ channel interactions, occurring between three adjacent CI electrodes.

Using three neighboring electrodes, we recorded eABR traces obtained 1) when stimulating independently from the three individual electrodes, or 2) when stimulating sequentially from the two flanking electrodes (E11 and E13), or 3) when stimulating sequentially from all three electrodes but varying the stimulation amplitude at E12. Our strategy was then mainly to compare the sum of eABR traces obtained from individual recordings in 1) to the sequential recording obtained in 3). The MIC was thus defined as the ratio between the sum of eABR wave V amplitudes observed on individual electrodes (1) and the amplitude of wave V observed in the sequential stimulation condition at different E12 stimulation levels (3). We hypothesized that the MIC would constitute a measure of electrode interaction, MIC values close to 1 reflecting low interactions (the sum of individual recordings and the sequential recording showing similar amplitudes). On the contrary, higher interaction levels would be indicated by larger MIC values, eABR wave V amplitudes being lower in the sequential stimulation condition than the sum of individual recordings from the same three electrodes. Neural channel interaction to stimulation on three adjacent electrodes was further assumed to increase with increasing stimulus intensity at the middle electrode.

Then the manuscript should standardize terminology, and a single term used always for each.  Does “sequential stimulation on E11 and E13” have the same meaning as “the dual stimulus condition” (Abstract)? Does “sequential stimulation of E11, E12, and E13” have the same meaning as “the triple stimulus condition” (Abstract)? The following sentence is especially confusing:  “Odd sequential” (OS) was obtained as the sum of eABR traces obtained in the dual stimulus condition and the single (sic) stimulus condition using E12”. As written, the paper is hard to understand. In the Abstract, the “Conclusions” should be more than the development of a paradigm.  What was observed from using that paradigm and how did it advance knowledge of electrode interactions?

Authors Answer:

We would like to thank Reviewer 2  for pointing to these issues related to the clarity of the abstract and manuscript and to the abstract structure. In order to answer these, we 1) re-wrote the abstract aiming at standardizing terminology to decrease potential confusions within the abstract and modified the conclusion section of the Abstract. We further modified the manuscript to reflect the standardization of terminology, we hope the new version of the manuscript is clearer and reproduce the abstract below for convenience:   

Abstract: Clinical performance with cochlear implants may be limited by electrical channel interactions. The present study investigates how electrically evoked Auditory Brainstem Responses (eABRs) could be used to measure local channel interactions. eABRs were recorded from 16 patients in response to three electrical stimulation conditions: 1) stimulation from single electrodes, 2) sequential stimulation on E11 and E13 and 3) sequential stimulation on E11, E12 and E13. Stimulation level was constant at 70% electrical dynamic range (EDR) on E11 and E13 and varied from 0 to 100% EDR on E12. Two main types of derived eABR traces were defined. Odd sequential (OS) was obtained as the sum of eABR traces obtained from the sequential stimulation on E11 and E13 and the single stimulation on E12. Continuous sequential (CS) was defined as the eABR trace recorded from the sequential stimulation on E11, E12 and E13. eABR wave III and V latency and amplitude were reduced in CS compared to OS, especially when stimulation level on E12 was 40, 70 and 100% EDR, reflecting increased channel interactions above 20% EDR stimulation. Using eABR can provide a detailed individual quantification of electrical interactions occurring at specific electrodes along the cochlear implant electrode array.   

The Methods say that eABR waveform peaks III and V were measured.  How were the peaks determined?  Two successive peaks are clear on Figure 3, but then where is peak IV? 

Authors Answer:

As stated in the Methods section of the original manuscript (section e.     Analysis of eABR traces and the monaural interaction component (MIC)): (…) eABR wave III and V peaks and valleys were identified visually based on their expected latencies. Wave III and V peak amplitudes were calculated as the difference (in µV) between the trough and its respective following positive peak for both waves. Peak latencies were calculated from the stimulus onset to the peak of the waveform.

It is usual that ABR waveforms and even more so eABRs, are actually recorded as 5-7 waves (see for example Robert M. Kliegman MD, in Nelson Textbook of Pediatrics, 2020). Actually, waves I, III, and V are obtained quite consistently in all age groups and in most stimulation conditions and individual patients. On the contrary, waves II and IV are somewhat more variable, both in terms of onset delay and of amplitude, hence they can disappear from grand-average recordings, even if they were observed in some individuals. Wave IV for example often occurs very close to Wave V on individual recordings, very often during the rise phase of wave V and tends to disappear when averaging individuals together (the more intense wave V is masking wave IV in group averages). This is why we concentrated on waves III and V that are more robustly identified in individual recordings. This is also why ‘only’ the main peaks of waves III and V can easily be identified on Figure 3.  

The Methods say “Stimulation rate was 47 Hz, and, in pulse trains, the inter-pulse interval was 20 uS”.  Please make it clear when pulse single pulses vs. pulse trains were used.  

Authors Answer:

We would like to thank Reviewer 2 for pointing to this confusing issue, actually throughout the experiment, only pulse trains were used to evoke eABR responses, we clarified this issue in the methods section, the corresponding paragraph now reads:

Stimulation pulses were the same as those used clinically i.e., charge-balanced, anodic first “pseudo-monophasic” pulses with a capacitive discharge phase. Pulse amplitude was fixed at 0.9 mA and pulse duration, corresponding to the duration of the leading anodic phase, was varied between 20 µs and 250 µs. Stimulation rate was 47 Hz, pulse onset asynchrony adjusted depending on pulse duration. eABR averaging involved 1,500 sweeps, recording duration was 10 milliseconds per sweep.

The Methods say, “The electrode insertion angles were estimated to be 171, 191, and 213 degrees – why are there 3 values?  Is this for 3 different patients?

Authors Answer:

The respective insertion angles were provided for electrodes E11, E12 and E13, hence the three different values, in the new version of the manuscript we rephrased the sentence to make this point clear. The modified paragraph now reads:

Electrode (E) number 11, 12 and 13, at an approximate cochlear position of 13.2, 14.4 and 15.6 mm from the round-window, were used for stimulation. The electrode insertion angles for these three electrodes were estimated to be 171°, 191° and 213° based on a medium-sized cochlea and using the model described in [31].

The Results begins (2nd sentence) with “the MIC increases”.  MIC is not defined and I cannot figure out what it is.

Authors Answer:

The definition of MIC was made clear in the early manuscript section added in response to comment 1 and is specified again in the Methods section of the revised manuscript. 

The Results are difficult to understand, because the first figure (Fig. 3) shows individual traces from one subject with the first paradigm, and then the second figure (Fig. 4) jumps to overall results of averaged traces from all subjects to a different paradigm. A more logical flow would make the reading less difficult.  

Authors Answer:

In the original manuscript, Figure 3 was thought as a graphical illustration of the individual traces observed in the different eABR manipulation conditions, and it was added to the Methods section. We understand however that this may be confusing, so we removed Figure from the Methods section to not include first results in the Methods section and respect a more ‘logical’ flow.

Figure 1: This figure is important but difficult to understand.  Is the x-axis time or distance?  If the stimuli are “sequential” then the stimulus on electrode E11 occurs before the stimulus on electrode E13 and the axis is time.  But the figure seems to be arranged as the x-axis is space and the stimuli occur simultaneously.  In the  legend, the following  sentence is unclear:  “The single and Triple combination shows an example depicting an increasing in dynamic range 137 (for 20 and 70% EDR). “  The figure’s “single” has a only one trace so how can it be an increasing dynamic range?  And the figure’s triple has 20% and 100% below the traces so how can that be 20% and 70%?  What do the “%” signs mean?

Authors Answer:

The electrode positions are depicted on Figure 1, so yes, the x-axis represents space. In cochlear implant stimulation, the stimuli are always “sequential”, however the delay between two electrodes (few tens of microseconds) is too small to have an impact on eABR measurements, this is why we neglect the fact that stimuli will occur with a delay on Figure 1, because from an eABR perspective it is not relevant. In order to reduce confusion, we decided to remove the non-relevant information about stimulation sequence from the Figure and the legend, in order not to mix time and space information in the same figure. For the second part of the commentary, concerning the legend, it is true that the original legend in the first version of the manuscript was misleading, we therefore corrected it in the new version of the manuscript, only the Triple condition is showing an example of the dynamic range growth from 20 to 100% for electrode E12, we thank Reviewer 2 for spotting this issue. The explanation concerning % EDR is just in the paragraph before the Figure and reproduced below, to account for individual variations in charge levels used for stimulation, intensities were always expressed in a % of the electrode electrical dynamic range between its lowest (T-level) and highest (C-level) stimulation intensity.

Due to the large variation of T- and most C-levels across participants, stimulation level was defined as the percentage of the electrical dynamic range (EDR) instead of the delivered charge. Stimulation level on E11 and E13 was always fixed at 70% EDR, and was changed on E12 to be 0, 20, 40, 70 or 100% EDR.

We hope that figure 1 and the accompanying legend are now clearer.

Figure 2: The right panel is confusing – do the lower traces result from E11 only and E12 only and E13 only being stimulated?

Authors Answer:

It is true that the right panel of Figure 2 was somewhat misleading, we decided to simplify Figure 2 and remove the right panel from its original version. We also updated the Figure’s legend accordingly.

Figure 3 legend: Should the legend read “visual designation” (instead of “visual assessment”)? And it is mentioned that there is “bump” at around 6 ms, but that is not visible in the traces.

Authors Answer:

Figure 3 and its legend were removed from the new version of the manuscript.

Discussion:  It is not clear that the first sentence is true – small waveform peaks can be greatly affected whereas large waveform peaks might be invariant. That sentence is not important for this paper and should be cut. The Discussion should begin with the “big-picture” items. 

Authors Answer:

The mentioned sentence was removed from the new version of the manuscript. 

One big-picture item that should be addressed in the Discussion is estimates of electrode interactions previously by others using different techniques (e.g., forward masking).  Do the author’s results indicate bigger or smaller than previous estimates?

Authors Answer:

The data presented in the current study do unfortunately not allow us to answer to this very interesting question raised by Reviewer 2. Because we did not perform any other measurement of electrical interaction sin the same participants as this was not the purpose of the present study, and because electrical interactions can be very variable from one patient to another one, or from  one cochlear implant system to another one (because of differences in number of electrodes, in stimulation mode etc.), it would only be possible to compare different interaction measures within the same patient group. It is not possible per se to compare the amount of interaction from our study with other studies from the literature as too many parameters vary and could play a role in observed differences. However, because the issue of comparing different methods to evaluate electrical interactions is really relevant and important, we added a mention concerning this possibility at the end of the conclusion paragraph. 
